# Early Deconditioning of Human Skeletal Muscles and the Effects of a Thigh Cuff Countermeasure

**DOI:** 10.3390/ijms222112064

**Published:** 2021-11-08

**Authors:** Théo Fovet, Corentin Guilhot, Laurence Stevens, Valérie Montel, Pierre Delobel, Rémi Roumanille, Michel-Yves Semporé, Damien Freyssenet, Guillaume Py, Thomas Brioche, Angèle Chopard

**Affiliations:** 1DMEM, INRAE, Université Montpellier, 34090 Montpellier, France; theo.fovet@umontpellier.fr (T.F.); corentin.guilhot@umontpellier.fr (C.G.); pierre.delobel@inrae.fr (P.D.); remi.roumanille@gmail.com (R.R.); guillaume.py@umontpellier.fr (G.P.); 2URePSSS Unité de Recherche Pluridisciplinaire Sport Santé Société, ULR 7369, Université Lille, F-69000 Lille, France; laurencestevens1963@gmail.com (L.S.); montel.valerie@univ-lille.fr (V.M.); 3Inter-University Laboratory of Human Movement Biology, University Jean Monnet Saint-Etienne, 42100 Saint-Étienne, France; mysempore@gmail.com (M.-Y.S.); damien.freyssenet@univ-st-etienne.fr (D.F.)

**Keywords:** spaceflight, muscle deconditioning, countermeasure, thigh cuff, dry immersion

## Abstract

Muscle deconditioning is a major consequence of a wide range of conditions from spaceflight to a sedentary lifestyle, and occurs as a result of muscle inactivity, leading to a rapid decrease in muscle strength, mass, and oxidative capacity. The early changes that appear in the first days of inactivity must be studied to determine effective methods for the prevention of muscle deconditioning. To evaluate the mechanisms of muscle early changes and the vascular effect of a thigh cuff, a five-day dry immersion (DI) experiment was conducted by the French Space Agency at the MEDES Space Clinic (Rangueil, Toulouse). Eighteen healthy males were recruited and divided into a control group and a thigh cuff group, who wore a thigh cuff at 30 mmHg. All participants underwent five days of DI. Prior to and at the end of the DI, the lower limb maximal strength was measured and muscle biopsies were collected from the vastus lateralis muscle. Five days of DI resulted in muscle deconditioning in both groups. The maximal voluntary isometric contraction of knee extension decreased significantly. The muscle fiber cross-sectional area decreased significantly by 21.8%, and the protein balance seems to be impaired, as shown by the reduced activation of the mTOR pathway. Measurements of skinned muscle fibers supported these results and potential changes in oxidative capacity were highlighted by a decrease in PGC1-α levels. The use of the thigh cuff did not prevent muscle deconditioning or impact muscle function. These results suggest that the major effects of muscle deconditioning occur during the first few days of inactivity, and countermeasures against muscle deconditioning should target this time period. These results are also relevant for the understanding of muscle weakness induced by muscle diseases, aging, and patients in intensive care.

## 1. Introduction

The next step in space exploration is long-duration space travel, which poses health challenges for astronauts. A microgravity environment causes several health problems during spaceflight, especially in the cardiovascular and musculoskeletal systems.

The absence of gravity induces a redistribution of body fluids, which results in a higher-than-normal blood concentration in the upper part of the body, leading to a decrease in left ventricle end-diastolic volume and stroke volume, arterial and venous resistance remodeling and enlargement, and heart deconditioning [1]. These changes result in facial oedema, discomfort, poor vascularization of the lower limbs, and vision problems [2,3,4]. Several cardiovascular countermeasures have been investigated to mitigate the fluid shift, including the use of lower body negative pressure (LBNP) [5]. In addition to LBNP, a thigh cuff countermeasure has been used by Russian astronauts due to its simplicity and the fact that it is not uncomfortable for the astronauts [1]. The thigh cuff is a compression sleeve that is worn on the upper thigh for a pre-determined length of time to limit the blood flow out of the thigh [6]. The thigh cuff partially compensates for cardiovascular changes induced by the microgravity environment during space explorations that result in vision impairment and discomfort [1,7]. Though the thigh cuff is widely used by astronauts, there is a lack of data regarding its effectiveness, the parameters associated with its use (such as pressure and wearing time), and its potential interactions with other systems. Most in-flight muscle studies have been conducted on astronauts who used the thigh cuff by default and, as a result, its effects on the muscle system remain unknown.

Skeletal muscle functions are also impaired by microgravity during spaceflight, and on earth associated with hypoactivity (sedentarity or patient in intensive care) and muscle diseases [8]. This phenomenon, termed muscle deconditioning, is associated with a severe and rapid loss of muscle strength. For example, more than 10% of leg extension strength may be lost after three days of hypoactivity [9]. Among several factors, muscle atrophy plays a major role in muscle deconditioning [10,11]. A protein imbalance (increased degradation and decreased synthesis) is one mechanism of muscle atrophy [8,12]. Muscle excitation–contraction coupling becomes deficient as a result of oxidative stress and inflammation [13,14]. The loss of strength is associated with the development of intercellular adipose tissue (IMAT), and an accumulation of IMAT is linked to muscle dysfunction [15]. Muscle deconditioning is also associated with an increase in muscle fatigability and metabolic changes. Hypoactivity induces phenotypical changes in the heavy chain of myosin in muscle fibers [16], changes in mitochondrial capacity and content, and the impairment of biogenesis, leading to low oxidative power [17,18,19,20,21]. 

The effects of hypoactivity-induced muscle deconditioning have been studied using several ground-based models, including the head-down tilt bedrest (HDTBR), which mimics unweighting and is a gold standard method for study muscle deconditioning [8,22,23]. In contrast, the dry immersion (DI) protocol [24] is a more radical yet appropriate model that reproduces most of the physiological effects that occur during spaceflight, including the centralization of body fluids, unloading of support, hypokinesia, and the lack of supporting structures under the body [25,26,27]. Spaceflight and DI models have shown that hypoactivity-induced muscle adaptations appear rapidly due to the high plasticity of muscle tissue [9,28]. Significant changes can be recorded as early as after two days of inactivity [29]. Many biochemical pathways become deactivated in the hours following muscle resting, initiating processes that lead to the loss of muscle mass and strength. A better understanding of these early and severe events is crucial to improve care in the clinic and during space flight to prevent muscle deconditioning.

## 2. Results

The participants’ characteristics are presented in Table 1. There were no significant differences between the groups at baseline. 

For each measurement and assessment included in the study, three sets of data were analysed: the control group (Ctrl), the thigh cuff group (Cuff), and a pooled group that included data from both groups. The pooled group was created due to the absence of differences between the Ctrl and Cuff groups. The experimental protocol is summarized in Figure 1.

### 2.1. Strength Measurements

After five days of DI, the knee extension MVC (maximal isometric voluntary contraction) decreased in both groups (Figure 2). The knee extension MVC of the Ctrl group decreased from 216.2 to 179.1 Nm (−17.17%; *p* < 0.001), and that of the Cuff group decreased from 218.7 to 199.1 Nm (−8.95%; *p* < 0.05). The knee extension MVC of the pooled group decreased from 217.4 to 188.3 Nm (−13.35%; *p* < 0.001). The knee flexion, ankle extension, and flexion MVC were not significantly different before and after DI in any group. The mean total strength of isolated slow muscle fibers in the pooled group decreased from 1.59 to 1.41 Nm after DI (*p* < 0.01) (Figure 2). The mean total strength of the slow muscle fibers in the Ctrl group decreased from 1.58 to 1.31 Nm (*p* < 0.01), and that of the Cuff group decreased from 1.6 to 1.52 Nm (*p* > 0.05). The mean total strength of isolated fast muscle fibers in the pooled group decreased from 1.59 to 1.4 Nm (*p* < 0.001). The mean total strength of the fast muscle fibers in the Ctrl group decreased from 1.6 to 1.36 Nm (*p* < 0.01), and that of the Cuff group decreased from 1.58 to 1.45 Nm (*p* > 0.05). The strength of slow fibers based on CSA decreased by 6% in the Ctrl group and by 4.32% in the Cuff group. The strength of fast fibers based on CSA decreased by 1.8% in the Ctrl group and 8.03% in the Cuff group. There were no significant differences in CSA before and after DI in any group. 

### 2.2. Atrophy Measurement

The CSA of muscle fibers decreased from 3778 µm^2^ to 2942 µm^2^ (−22.1%; *p* < 0.05) in the Ctrl group, from 3971 µm^2^ to 3113 µm^2^ (−21.6%; *p* < 0.05) in the Cuff group, and from 3867 µm^2^ to 3021 µm^2^ (−21.8%; *p* < 0.001) in the pooled group (Figure 3). The CSA of the slow muscle fibers decreased from 3724 µm^2^ to 2920 µm^2^ (−21.5%; *p* < 0.01) in the Ctrl group, from 3907 µm^2^ to 2881 µm^2^ (−26.2%; *p* < 0.01) in the Cuff group, and from 3808 µm^2^ to 2851 µm^2^ (−25.12%; *p* < 0.001) in the pooled group. The CSA of the fast muscle fibers decreased from 3800 µm^2^ to 2915 µm^2^ (−23.27%; *p* < 0.01) in the Ctrl group, from 4000 µm^2^ to 3237 µm^2^ (−19.07%; *p* < 0.05) in the Cuff group, and from 3892 µm^2^ to 3054 µm^2^ (−21.54%; *p* < 0.01) in the pooled group. 

### 2.3. Protein Balance Assessment

To characterize the impact of 5 days of DI on protein balance, we evaluated markers of main protein synthesis (Figure 4) and degradation pathways (Figure 5 and Figure 6). In the protein synthesis pathway, mTOR phosphorylation (ser^2448^) decreased by 24.24% (*p* < 0.05) and 4EBP1 phosphorylation (ser^65^) decreased by 29.22% (*p* < 0.01) in the pooled group (Figure 4). Changes in protein synthesis were not significant in the Ctrl and Cuff groups. The phosphorylation of RPS6 (ser^240/244^), another target of mTOR, remained unchanged. UPS E3 ligase Murf1 expression was upregulated by 29.81% in the pooled group (*p* < 0.01) and by 44.63% (*p* < 0.01) in the Ctrl group (Figure 5). The protein levels of Mafbx, another main E3 ligase involved in atrophy, did not change in either group. The total ubiquitinated protein content increased by 16.36% (*p* < 0.05) in the pooled group. Phosphorylation of the atrophy initiation marker ULK1 (ser^555^) increased by 56.7% (*p* < 0.05) in the pooled group (Figure 6). Atg7 protein expression increased by 39.67% (*p* < 0.01) in the pooled group and by 42.8% (*p* < 0.05) in the Cuff group. The expression of the p62 protein aggregate remained unchanged in both groups. Calpain 1 protein expression was upregulated by 18.68% (*p* < 0.01) in the pooled group and by 31.88% (*p* < 0.01) in the Cuff group (Figure 6). Calpain 2 protein expression remained unchanged in all groups. 

### 2.4. Muscle Typology and Oxidative Capacity

Prior to DI, 48.83% of muscle fibers in the Ctrl group were slow fibers (type 1), and after DI, 45.12% of muscle fibers in the Ctrl group were slow fibers. In the Cuff group, 44.45% of muscle fibers were slow fibers prior to DI, and 47.83% were slow fibers after DI (Figure 3). The PGC1-α protein expression decreased by 37.8% (*p* < 0.001) in the pooled group, by 28.75% (*p* < 0.05) in the Ctrl group, and by 46.03% (*p* < 0.001) in the Cuff group (Figure 7). The expression of mitochondrial fusion and fission cycle markers Mfn2 and Fis1 did not change with respect to the expression of respiratory chain markers cytochrome C and COXIV. The protein levels of mitochondrial activity marker citrate synthase and mitophagy marker parkin were unchanged in all groups. 

### 2.5. Inflammation

The protein levels of TNF-α and IL-1β were unchanged in the Ctrl and Cuff groups (Figure 8). 

### 2.6. Oxidative Damage and Antioxidant Defences

Oxidative stress variations were evaluated through the double aspect of damage in the cells and antioxidant cell defence adaptations. The 4HNE levels increased by 35.09% (*p* < 0.05) in the pooled group (Figure 9). In parallel, the carbonylated protein levels remained stable in the Ctrl and Cuff groups. For antioxidant cell defence adaptations, Gpx1 protein expression increased by 37.59% (*p* < 0.01) in the pooled group; however, the change in Gpx1 protein expression was not significant in the Ctrl or CT groups. SOD1, SOD2, and catalase protein levels were not significantly different in any group.

## 3. Discussion

This study investigated the early mechanisms of muscle deconditioning following five days of DI and evaluated the effects of a 30–50 mmHg thigh cuff countermeasure in the context of space exploration and clinical research. After five days of DI, participants were found to have a loss in muscle strength, muscle fiber atrophy, impaired protein balance, and decreased oxidative capacity, regardless of the use of a thigh cuff. In contrast, measurements on cardiovascular and ophthalmological function carried out by our collaborators have shown a beneficial effect of the thigh cuff to prevent the adverse effects of body fluid redistribution induced by 5 days of DI [30,31,32].

### 3.1. Five Days of DI Induced a Rapid Loss of Muscle Strength 

Strength loss is the best functional marker of muscle deconditioning, and its measurement provides an assessment of the validity of a study’s methods and quantifies the severity of muscle impairment. In this study, five days of DI induced a significant loss of strength (Figure 2). While knee and ankle extension MCV decreased, ankle flexion MVC did not change, which may be due to a higher glycolytic activity of the involved muscles. To go further, strength was measured on single muscle fiber of *vastus lateralis.* The same results than knee extension MVC, where *vastus lateralis* is involved, was found. Together, these results suggest that the thigh cuff did not significantly affect the loss of strength during DI. The results of this study are consistent with previous results related to ground-based model experiments. Demangel et al. showed a knee extension MVC loss of 9.1% after three days of DI [9]. After two months of bedrest, knee extension strength was reduced by 32% and ankle extension MVC was reduced by 16% [21]. Results from different time points allow for a better understanding of muscle deconditioning that occurs in these ground-based models. Hypoactivity-induced strength loss is a result of several factors, including muscle atrophy [10,11].

### 3.2. Five Days of DI Induced Significant Muscle Fiber Atrophy

To explain the lower limb loss of strength after 5-day DI, we investigated *vastus lateralis* muscle atrophy (Figure 3). The CSA of the muscle fibers of the VL decreased after five days of DI. Pooled subjects’ CSA showed 21.8% of atrophy after DI. We then measured muscle atrophy for each type of fiber regarding their different molecular responses to hypoactivity. Atrophy was observed in both types of muscle fibers in this study, despite differences in molecular responses to hypoactivity between the fiber types [17,21,33]. As no significant differences in atrophy were observed between the Ctrl and Cuff groups, the thigh cuff did not prevent muscle mass loss in this study. These results may be due to the very low pressure applied by the thigh cuff. Such effects have been reported in BFR-type experiments (blood flow restriction), but these studies use much higher pressures than this study (more than 200 mmHg for the BFR), and are not comparable [34]. The results of this study are consistent with those of other ground-based models: Demangel et al. reported VL muscle atrophy of 10% after three days of DI [9]; Edgerton et al. reported an 11% and 16% decrease in MyHc1 fiber CSA and a 24% and 36% decrease in MyHc2 fiber CSA after five and 11 days of spaceflight, respectively [35]. The DI model may have resulted in more atrophy as the astronauts were required to perform exercises during spaceflight. The MVC and muscle fiber CSA results in this study indicate a decrease in muscle strength and muscle atrophy, which is not consistent with previous studies [10]. These differences suggest that atrophy occurs at an accelerated pace at the beginning (during the first few days) of hypoactivity and slows with longer exposure to hypoactivity to limit muscle mass loss. 

### 3.3. Five Days of DI Impaired the Muscle Protein Balance

Hypoactivity is known to significantly alter the protein balance by decreasing protein synthesis and increasing protein degradation, leading to a negative protein balance, which results in the induction of the atrophy process [12]. Protein synthesis is mainly regulated by the insulin growth factor 1-phosphoinositide 3 kinase-protein kinase B-mammalian target of rapamycin (IGF1-PI3K-Akt-mTOR) pathway [36,37]. After five days of DI, mTOR pathway markers showed a low level of activation, suggesting a decrease in protein synthesis (Figure 4). These results are in accordance with the drop in protein synthesis occurred as early as 24 h after the beginning of muscle inactivity and the inhibition rate remained high during the first few days [38]. In contrast, proteolytic systems involved in protein degradation during hypoactivity (such as UPS and autophagy) were upregulated [39]. After five days of DI, the protein content and/or activation of the main markers of these pathways were increased (Figure 5 and Figure 6). Gustafsson et al. reported an increase in Murf1 E3 ligase after three days of unilateral leg unloading [40]. Similarly, studies using the bedrest model showed an increase in autophagic ATG and markers after 24 days [41] and an increase in Murf1 protein content after 60 days [42]. Taken together, these results indicate that protein degradation upregulation begins early during hypoactivity, as protein synthesis slows within three days of hypoactivity and persists over time. Mechanistically, in the microgravity context, the inhibition of the mTOR pathway leads to the activation of FoxO3 and ULK1, both of which allow the initiation of autophagy and the transcription of Mafbx and MurF1 [43]. Together, the drop-in protein synthesis rate and upregulation of protein degradation contribute to muscle atrophy. These results indicate that the thigh cuff has no impact on protein balance regulation. 

### 3.4. Hypoactivity Affects Muscle Strength in Several Ways

The decrease in muscle strength that occurs after periods of DI or exposure to microgravity cannot be fully attributed to muscle atrophy. Impairments in neural drive and excitation–contraction coupling have been suggested as cofactors [13,14]. A decrease in motoneuron alpha activation capacity has been reported in a ground-based (including bedrest and hindlimb unloading) and spaceflight models [44]. In addition, age-induced muscle deconditioning is associated with an increase in denervated muscle fibers, contributing to a decrease in strength [45]. Demangel et al. reported that three days of DI results in a small but significant increase in denervated muscle fiber markers [9]. More studies are needed to identify the initiation of the denervation process during severe muscle disuse. Excitation–contraction coupling is impaired by the appearance of inactivity-induced oxidative stress that results in the oxidation of ryanodine receptors and SERCA (sarco/endoplasmic reticulum Ca^2+^-ATPase) and inhibits membrane calcium ATPase activity [14,46,47]. This oxidation impairs the calcium cycle in the muscle excitation–contraction coupling, subsequently impairing muscle strength, which may favour activation of calcium-dependent proteases (calpains) and the pro-apoptotic caspase pathway [14]. In this study, the protein levels of calpain 1 and 4HNE were upregulated after five days of DI in the pooled group. These results highlight the significant and early loss of muscle strength and are consistent with muscle atrophy and contractile element degradation function of calpain [48,49].

### 3.5. Five Days of DI Seems to Initiate a Decrease in Muscle Oxidative Capacity

Muscle deconditioning is secondary to an increase in muscle fatigability and metabolic changes caused by a loss of muscle oxidative capacity [8,50]. One explanation for the decrease in muscle oxidative capacity is a shift towards glycolytic or type 2 muscle fibers. Although five days of DI did not alter the fiber type distribution of either group in this study, a decrease in mitochondrial biogenesis (observed as the downregulation of PGC1-α protein expression) was noted in the pooled group (Figure 7). However, mitochondrial fission and fusion dynamics, measured via FIS1 and MFN2 protein content, respectively, were not affected in either group in this study. The mitochondrial citrate synthase (CS) content and cytochrome C and COX IV expressions were also not affected in either group. These results suggest that the mechanisms leading to a loss of oxidative capacity have been initiated but are not yet fully effective after five days of DI and that the thigh cuff does not affect oxidative capacity. In a previous study, Demangel et al. showed that three days of DI resulted in a decrease of PGC1-α, FIS1, MFN2, citrate synthase, and COXIV levels, confirming that the adaptations or timing of the adaptations differ between individuals during the first days of inactivity [9]. Changes in the atrophic process appear earlier than changes in the metabolic shift. Longer periods of severe muscle disuse, as in bedrest, result in changes that are consistent with the significant decrease in mitochondrial content measured in spaceflight [17,18,19,20,21]. The decrease in aerobic capacity significantly affects most of the postural muscles [51], including the soleus and spinal muscles [52,53,54], as opposed to fast muscles such as the deltoid, which shows minimal effects of hypoactivity at the mitochondrial level [55]. These mitochondrial impairments enhance the muscle atrophy process, leading to muscle deconditioning [56].

### 3.6. Five Days of DI Does Not Increase Intra-Muscular Inflammation Markers 

Inflammation pathways are frequently identified as clinical outcomes during the first stage of muscle deconditioning, especially in elderly individuals [57,58]. However, the durability of these pathways over time is unclear [59]. After five days of DI, the muscle content of TNF-α did not change in any group in this study. While the IL-1β content trended towards upregulation in both groups, the changes were not significant. These results indicate that more studies regarding inflammatory processes and responses during periods of inactivity are needed to better understand the individual factors leading to these important variations.

### 3.7. Five Days of DI Resulted in an Increase in Muscle Oxidative Damage Production 

Oxidative stress is a key regulator of several pathways involved in muscle deconditioning and is mainly linked to muscle strength loss and atrophy during hypoactivity. In excessive quantities, reactive oxygen and nitric species RONS impair protein balance, macromolecules, and organelles, cause cell damage, and upregulate the apoptosis pathway [14]. After five days of DI, the level of oxidative damage to proteins (measured as the carbonylated protein level) remains unchanged, though 4HNE expression (oxidative damage to lipids) increased in the pooled group in this study. Cells use several enzymatic and non-enzymatic antioxidant defences to counteract RONS. In this study, the Gpx1 protein content was upregulated in the pooled group. These results suggest an increase in the RONS levels during the first stage of DI and are consistent with the results of other studies [60,61]. Although oxidative stress levels are specific to individuals, RONS production may be reduced through nutritional or physical-based countermeasures that can restore a harmonious REDOX balance and prevent numerous RONS-induced pathways that contribute to muscle deconditioning. Several biochemical pathways are deactivated in the hours following muscle resting, initiating processes that lead to the loss of muscle mass, strength, and oxidative capacity. A better understanding of these early events will allow for their prevention. While several muscle deconditioning countermeasures have been investigated, the most efficient countermeasure is physical activity. More research regarding countermeasures is needed to develop a set of countermeasures to prevent or inhibit the effects of weightlessness in a more global manner. Authors should discuss the results and how they can be interpreted from the perspective of previous studies and of the working hypotheses. The findings and their implications should be discussed in the broadest context possible. Future research directions may also be highlighted.

## 4. Materials and Methods

### 4.1. Participants

This study was conducted at the MEDES Space Clinic in Toulouse, France, from 19 November 2018 to 23 March 2019. Twenty healthy males were recruited, though two participants withdrew before the study began for unrelated reasons. Therefore, 18 participants were evenly and randomly divided into the control and cuff groups two days prior to the DI. All participants provided written informed consent for their participation in the study. The study followed the principles of the Declaration of Helsinki and was approved by the local ethics committee (CPP Est III: 2 October 2018 n° ID RCB 2018-A01470-55) and French Health Authorities (National Agency for the Safety of Medicines and Health Products on 13 August 2018). The study was also registered at ClinicalTrials.gov (NCT03915457). The participants had no history or physical signs of neuromuscular disorders, were non-smokers, and did not take drugs or medications. The participants arrived at the space centre in the evening five days prior to the start of DI and left the morning two days after DI was completed (Figure 1). Ambulatory baseline measurements were recorded for four days prior to DI, then the participants completed five days of DI followed by two days of ambulatory recovery. A pre-immersion control muscle biopsy was collected and the resting metabolic rate was measured. Participants in the cuff group wore thigh cuffs at 30–50 mmHg from 10 am to 6 pm on the first day of DI and from 8 am to 6 pm thereafter. Individual adjustments were made for each participant based on calf plethysmography measurements obtained with the participant in the supine position on the second day of DI. On the first day, thigh cuffs were placed immediately prior to the onset of immersion at 10 am. DI was conducted according to previously-described, strict methodology [62]. Two participants, one control and one cuff, underwent DI simultaneously in two separate baths in the same room. A thermoneutral water temperature was continuously maintained (33 ± 0.5 °C). The light-off period was set at 23:00–07:00. Daily hygiene, weighing, and specific measurements required extraction from the bath. During these out-of-bath periods, the subjects maintained the −6° head-down position. The total out-of-bath supine time during the 120-h immersion period was 9.7 ± 1.3 h, and the daily out-of-bath time was 1.1 ± 0.6 h. On the last day of DI, the out-of-bath time was 5.3 ± 1.1 h due to the muscle biopsy procedure and MRI exam. During DI, the participants remained immersed in a supine position for all activities and were continuously observed via video monitoring. Body weight, blood pressure, heart rate, and tympanic body temperature were measured daily. Water intake throughout the protocol was ad libitum, though maintained between 35–60 mL/kg/day and was measured. The menu was identical for all participants for each day of the study, and dietary intake was individually tailored and controlled.

### 4.2. Strength Measurements

To determine the strength of the participants’ lower limb muscles, the maximal voluntary isometric contraction (MVC) strength was measured using a Con-Trex device (Physiomed; Schnaittach, Germany). The strengths of the knee and ankle flexors and extensor muscle groups were measured on the left leg. MVC was determined at 80° extension of the knee and 0° extension of the ankle. Measurements were obtained four days prior to the start of DI and just before the end of DI (during the restart of the upright posture) (Figure 1). Each participant was familiarised with the equipment and standardised protocol before the measurements were obtained. During the assessments, participants were seated and firmly attached to the chair of the ConTrex device to avoid movement. The protocol was similar for each muscle group; after a short warm-up in a neutral position, a series of measurements were recorded with a 30-s recovery interval. Each series consisted of an extension movement followed by an isometric contraction and a flexion movement followed by an isometric contraction. Each contraction was maintained for 5–7 s, and a two-minute recovery period was permitted after every three sets of measurements. The total duration of the test was 15 min per agonist/antagonist muscle group pairs. To determine the MVC, the maximum strength level (Nm) achieved during the test was recorded.

### 4.3. Muscle Biopsy

A skeletal muscle biopsy was conducted on the right vastus lateralis muscle (VL) prior to the start of DI (pre-DI) and before re-ambulation on the last day of DI (post-DI) according to a well-established method using a 5 mm Bergström biopsy needle under sterile conditions and local anaesthesia (1% lidocaine) [63]. The pre- and post-DI biopsies were obtained from the same leg from areas as close together as possible. The biopsy was then separated into several pieces for the following analyses. For the immunohistological analyses, one piece of the biopsy tissue was immediately embedded in small silicone casts filled with a cryoprotector (OCT, Sakura Finetek, Torrance, CA, USA), frozen in isopentane-cooled nitrogen, and stored at −80 °C. The remaining biopsy tissue was rapidly frozen in liquid nitrogen and stored at −80 °C for mRNA and protein content quantification.

### 4.4. Preparation of Biopsies for Skinned Fiber Experiments

Immediately after sampling, a piece of biopsy tissue oriented in the longitudinal axis was chemically skinned for contractile experiments on single muscle fibers. The skinning procedure was based on Ca^2+^ chelation by EGTA (egtazic acid), which permeabilizes the sarcolemma and transverse tubular membranes. The EGTA skinning solution (see solutions and reagents section below) was applied for 24 h at 4 °C. The skinned biopsies were then stored at −20 °C in a 50:50 glycerol-skinning solution (storage solution) into which the protease inhibitor leupeptin was added (10 µg/mL) to prevent protein degradation. For each experiment, a 2- to 2.5-mm single-fiber segment was isolated from the skinned biopsy. Silk was tied to each extremity, and the fiber was mounted in an experimental chamber with a bath temperature of 19 ± 1 °C, initially filled with relaxing solution (R) with constant stirring. The fiber was held at one end using small forceps and at the other end by a clamp connected to a strain gauge (force transducer Fort 10; sensitivity 10 V/g). 

### 4.5. Skinned Fiber Solutions and Reagents 

All reagents were purchased from Sigma-Aldrich (St. Louis, MO, USA). The composition of all solutions was based on previously described protocols [64] and calculated using the Fabiato program [65]. The pH was adjusted to 7.0 ± 0.02, and the final ionic strength was adjusted to 200 mM. ATP (2.5 mM) was added to each solution. The skinning solution was composed of 10 mM MOPS, 170 mM potassium propionate, 2.5 mM magnesium acetate, and 5 mM K_2_EGTA. The washing solution (W solution) was composed of 10 mM MOPS, 185 mM potassium propionate, 2.5 mM magnesium acetate, 10 mM phosphocreatine, and the relaxing solution was identical to the skinning solution (R solution). The Ca^2+^ activating solution consisted of W solution plus a concentration of free Ca^2+^ from CaCO_3_, buffered with EGTA, and added in increments to obtain the final pCa values. The Sr^2+^ activating solution consisted of W solution plus a concentration of free Sr^2+^ from SrCl_2_, buffered with EGTA, added in increments to obtain the final pSr values.

### 4.6. Skinned Fiber Type and Force Measurements

Twenty fibers (10 slow and 10 fast fibers) were selected per muscle according to the functional phenotype determined using the calcium/strontium test [66]. After the fiber was mounted in R solution, the diameter and sarcomere length were measured. Then, the fiber was bathed in W solution, which removed all EGTA left from the R solution. The fiber was then activated using pCa 4.2 solution until maximal tension (P_0_) at the plateau was achieved. Then, the fiber was relaxed in R solution followed by a fiber identification test [67] based on the fact that fast muscle fibers are less sensitive to Sr^2+^ than slow fibers [68]. Thus, the fiber was activated with a submaximal pSr 5.0 solution, followed by the application of a pSr 3.4 solution, resulting in maximal Sr tension. If the ratio pSr 5.0/pSr 3.4 was 90–100%, the fiber was considered to be slow type and fast fibers had a pSr 5.0/pSr 3.4 ratio of 0–10%.

### 4.7. Immunohistological Analysis

To evaluate VL atrophy, the cross-sectional area (CSA) of the muscle fiber was measured before and after DI on transverse cryosections. Serial transverse cross-sections (10 μm thick) of VL samples were cut using a cryostat at −25 °C. The sections were dried and fixed for 10 min in acetone before being washed in phosphate-buffered saline (PBS), blocked, and permeabilized with 0.1% Triton-X100 and 20% horse serum. The sections were incubated with anti-MyHC antibodies (Table 2) for 1 h at 37 °C, followed by an incubation with the corresponding secondary antibody for 1 h at 37 °C (Table 2). The sections were then incubated with anti-laminin (Table 2) for 1 h at 37 °C, rinsed in PBS, and incubated with the corresponding secondary antibody for 1 h at 37 °C (Table 2). Nuclei were labelled using Hoesch staining solution (1/1000) for 30 s. The sections were then set up with permafluor and dried overnight at room temperature. Pictures were obtained using a ZEISS AxioScan (INM, Montpellier) at a 20× magnification with a focus on the DAPI-stained nuclei. The fiber size and typology were analysed using Image J software (version 1.46).

### 4.8. Western Blot Analysis

Muscle samples were homogenised in 10 volumes of lysis buffer (50 mM Tris–HCl (pH 7.5), 150 mM NaCl, 1 mM egtazic acid, 1 mM EDTA, 100 mM NaF, 5 mM Na_3_VO_4_, 1% Triton X100, 1% sodium dodecyl sulphate (SDS), 40 mM β-glycerophosphate, and a protease inhibitor mixture (P8340; Sigma-Aldrich, Saint-Louis, MO, USA) and centrifuged at 10,000× *g* for 10 min (4 °C). Fifty micrograms of protein extract were loaded into stain-free, 4–20% precast gels (4568095; Bio-Rad, Hercules, CA, USA) before electrophoretic transfer onto nitrocellulose membranes (Trans-Blot Turbo Blotting System; Bio-Rad). After the transfer, stain free staining was realized as the loading control. The membranes were blocked with 50 mM Tris–HCl (pH 7.5), 150 mM NaCl, and 0.1% Tween 20 (Tris-buffered saline-T) containing 5% skim milk or BSA and incubated overnight at 4 °C with primary antibodies (Table 2). The membranes were then incubated for 1 h with a secondary antibody (Table 2). Immunoblots were detected using a Pierce ECL kit (32106; Thermo Fisher Scientific, Waltham, MA, USA), and proteins were visualised using enhanced chemiluminescence with the ChemiDoc Touch Imaging System (Bio-Rad) and quantified with Image Lab™ Touch Software (version 5.2.1). Stain-free technology was used as the loading control. 

**Table 2 ijms-22-12064-t002:** Antibody table.

Antibody	Reference	Commercial	Dilution
4-HNE	ab46545	Abcam	1:2000
p62	ab56416	Abcam	1:1000
p-4EBP1	9451S	Cell signaling	1:1000
4EBP1	9644S	Cell signaling	1:1000
Atg7	8558S	Cell signaling	1:1000
Gpx1	3206S	Cell signaling	1:1000
Calpain 1	2556S	Cell signaling	1:1000
Calpain 2	2539S	Cell signaling	1:1000
p-Rps6	5364S	Cell signaling	1:1000
Rps6	3944S	Cell signaling	1:1000
p-mTOR	5536S	Cell signaling	1:1000
mTOR	2983S	Cell signaling	1:1000
p-ULK1	5869S	Cell signaling	1:1000
ULK1	8054S	Cell signaling	1:1000
Anti-mouse HRP	7076	Cell signaling	1:5000
Anti-Rabbit HRP	7074	Cell signaling	1:5000
Anti-laminin	L9393	Sigma Aldrich	1:200
Anti-MyHC 1	BA-D5	DSHB	1:10
Anti-MyHC 2a	SC-71	DSHB	1:10
Catalase	110704	Genetex	1:1000
SOD1	100554	Genetex	1:1000
SOD2	116093	Genetex	1:1000
Anti-Rabbit Alexafluor 588	A11036	Invitrogen	1:800
Anti-Mouse Alexafluor 488	A21121	Invitrogen	1:800
PGC1-α	AB3242	Millipore	1:1000
Ub	sc-8017	Santa-Cruz	1:200
Mafbx	sc-33782	Santa-Cruz	1:200
Murf1	sc-27642	Santa-Cruz	1:200
Cytochrome C	sc-13560	Santa-Cruz	1:200
Citrathe synthase	sc-390693	Santa-Cruz	1:200
COX IV	sc-69360	Santa-Cruz	1:200
Fis1	sc-98900	Santa-Cruz	1:200
Mfn2	sc-515647	Santa-Cruz	1:200
TNF-α	sc-52746	Santa-Cruz	1:200
IL-1β	sc-7884	Santa-Cruz	1:200
Parkin	Sc-32282	Santa-Cruz	1:200
Anti-Goat HRP	sc-2953	Santa-Cruz	1:4000
Anti-MyHC 2	M4276	Sigma-Aldrich	1:200

### 4.9. Carbonylated Protein

An OxyBlot Protein Oxidation Detection Kit (Millipore, Burlington, MA, USA) was used to detect carbonylated protein. Protein samples were denatured with 12% SDS (Sodium Dodecyl Sulfate) at a final concentration of 6% SDS. The samples were then derivatized using 2,4-dinitrophenylhydrazine (DNPH) and incubated for 15 min at room temperature. The reaction was stopped using a neutralisation solution before being loaded into a 4–20% precast gel (5678094; Bio-Rad) before electrophoretic transfer onto nitrocellulose membranes (Trans-Blot Turbo Blotting System; Bio-Rad). Membranes were washed in phosphate-buffered saline with 1% of Tween 20 (PBS-T) and incubated with primary antibody diluted in blocking solution buffer (1:150) at room temperature for 1 h. The membranes were washed again in PBS-T and incubated with a secondary antibody diluted in blocking solution buffer (1:300) at room temperature for 1 h. Finally, membranes were washed and incubated for five minutes using the Pierce ECL kit (32106; Thermo Fisher Scientific), and proteins were visualised using enhanced chemiluminescence with the ChemiDoc Touch Imaging System and quantified with Image Lab™ Touch Software (version 5.2.1). Ponceau staining was used as the loading control. 

### 4.10. Statistical Analyses

All values are expressed as mean ± standard deviation. Differences between pre-DI and post-DI and between groups were evaluated using two-way analysis of variance (ANOVA) matched-pairs with Tuckey HSD post hoc or Friedman ANOVA when the data deviated from a normal distribution, as determined using the Shapiro–Wilk normality test. Statistical analyses were performed using Statistical Software (version 7.1) and graphs were created using GraphPad Prism4 software (San Diego, CA, USA). Statistical significance was set at *p* < 0.05.

## 5. Conclusions

Understanding the precise mechanisms of muscle deconditioning and developing countermeasures is an important health challenge for human space exploration as well as for clinical diseases and aging on Earth. In this study, healthy participants underwent five days of DI with or without a thigh cuff to determine the effects of the cuff on cardiovascular and muscle disorders. This short immersion resulted in a loss of muscle strength and mass and decreased oxidative capacity. These effects were verified by the observation of the upregulation of oxidative stress, a protein imbalance, and the impairment of mitochondrial biogenesis. Despite the results on vascular and ophthalmic function in other studies, the use of a thigh cuff had no effect on muscle function. These results suggest that several deconditioning processes are initiated within the first five days of hypoactivity. Therefore, nutritional and physical-based countermeasures that limit the atrophying process should be started as early as the first two days of inactivity. This experimental dry immersion model does not mimic the real life conditions of astronauts. However, it is one of the most reliable protocols for this type of research and is a unique opportunity to develop countermeasures of muscle deconditioning on healthy people.

## Figures and Tables

**Figure 1 ijms-22-12064-f001:**
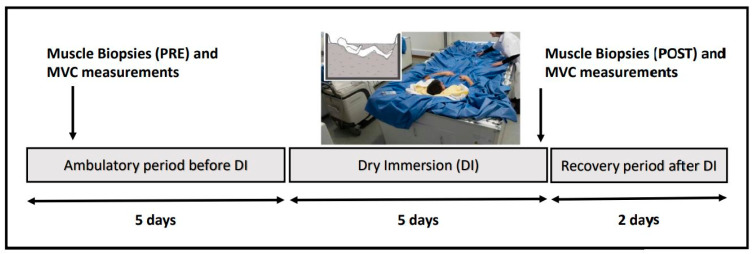
Experimental protocol. Subjects arrived in the evening 5 days before DI and left 2 days after DI in the morning. The experimental protocol included four days of ambulatory baseline measurements before immersion, five days of dry immersion, and two days of ambulatory recovery. Baseline and post measurement included muscle biopsy of the vastus lateralis and maximal voluntary contraction measurement (MVC). Subjects randomized to cuff group wore the thigh cuffs during the 5 days of DI, from 10 h to 18 h the first day and from 8 h to 18 h the other as at counter-pressure of 30–50 mmHg.

**Figure 2 ijms-22-12064-f002:**
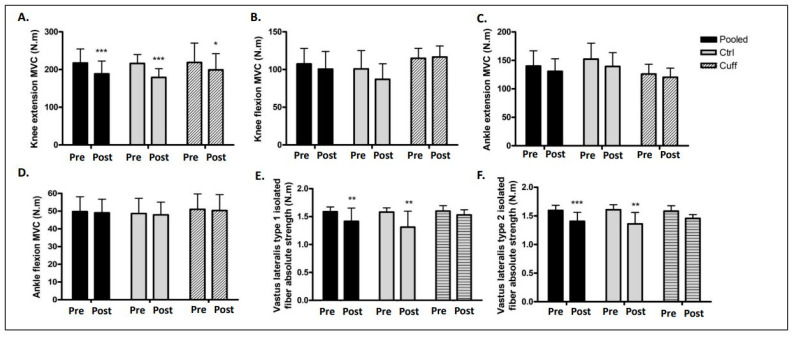
Effects of 5 days of DI on muscle and fiber strength. Muscle maximal isometric voluntary contraction access by isocinetic dynamometer (N.m) expressed in both groups and pooled subjects. (**A**) Knee extension measurement. (**B**) Knee flexion measurement. (**C**) Ankle extension measurement. (**D**) Ankle flexion measurement. Isolated muscle fiber strength production by fiber type. (**E**). Type 1 isolated muscle fiber contraction. (**F**). Type 2 isolated muscle fiber contraction. *: different from pre value in the same group. * *p* < 0.05; ** *p* < 0.01; *** *p* < 0.001.

**Figure 3 ijms-22-12064-f003:**
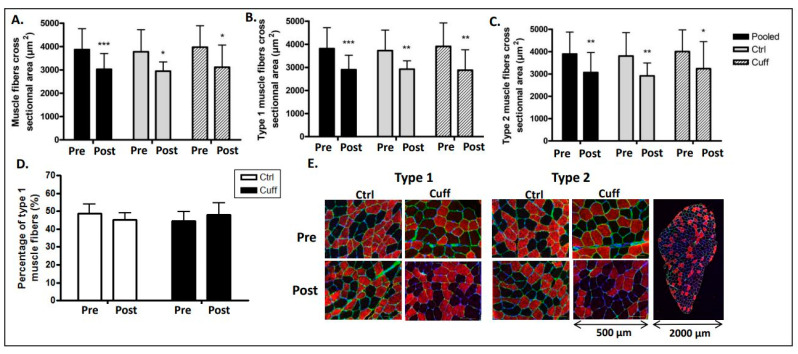
Effect of 5 days of DI on muscle fiber cross-sectional areas and fiber type distribution. Muscle atrophy measured with muscle fiber cross sectional area (CSA) in µm^2^ and muscle fiber typology. (**A**) Muscle fiber CSA on immunostained cryosections. (**B**) MyHC1 muscle fiber’s CSA. (**C**) MyHC2 muscle fiber’s CSA. (**D**) Typology measurement express by percentage of type 1 muscle fiber on immunostained cryosections. (**E**). Analyzed pictures of muscle samples labelled with anti-lamnin (green), anti-MyHC1 or MyHC2 (red) and hoesch (blue). * different from Pre value in the same group. * *p* < 0.05; ** *p* < 0.01; *** *p* < 0.001.

**Figure 4 ijms-22-12064-f004:**
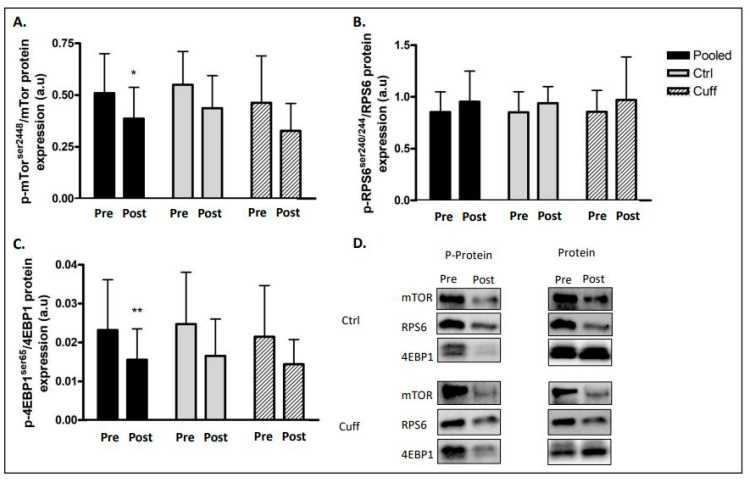
Effect of 5 days of DI on protein synthesis marker. Protein synthesis marker activation access by western blot analysis. (**A**) p-mTor^ser2448^/mTor protein ratio. (**B**) p-RPS6^ser240/244^/RPS6 protein ratio. (**C**) p-4EBP1^ser65^/4EBP1 protein ratio. (**D**) Western blot picture analyzed by groups. * different from pre value in the same group. * *p* < 0.05; ** *p* < 0.01.

**Figure 5 ijms-22-12064-f005:**
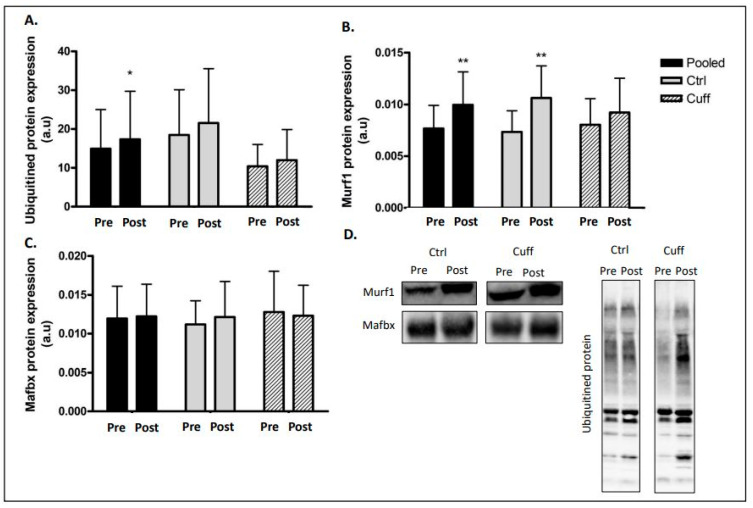
Effect of 5 days of DI on ubiquitin proteasome pathway. UPS marker activation access by western blot analysis. (**A**) Ubiquitinated protein profile expression. (**B**) E3 ligase Murf1 protein expression. (**C**) E3 ligase Mafbx (Atrogin1) protein expression. (**D**) Western blot picture analyzed by groups. * different from pre value in the same group. * *p* < 0.05; ** *p* < 0.01.

**Figure 6 ijms-22-12064-f006:**
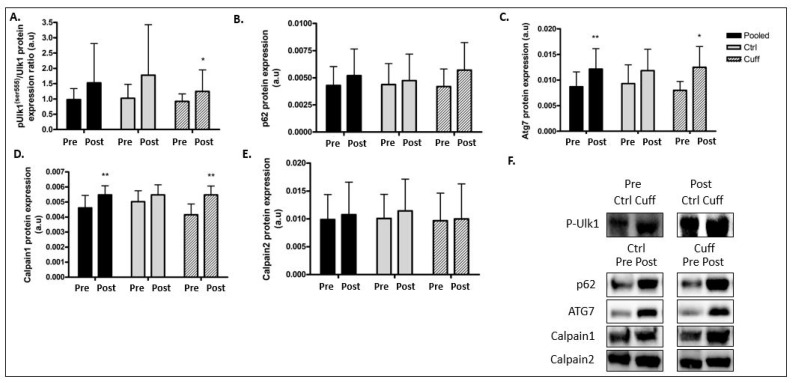
Effect of 5 days of DI on autophagic and calpain markers. Autophagy markers activation access by western blot analysis. (**A**) pUlk1^ser757^/Ulk1 protein ratio. (**B**) p62 protein expression. (**C**) ATG7 protein expression. (**D**) Calpain1 protein expression measurement. (**E**) Calpain2 protein expression measurement (**F**) Western blot picture analyzed by groups. * different from pre value in the same group. * *p* < 0.05; ** *p* < 0.01.

**Figure 7 ijms-22-12064-f007:**
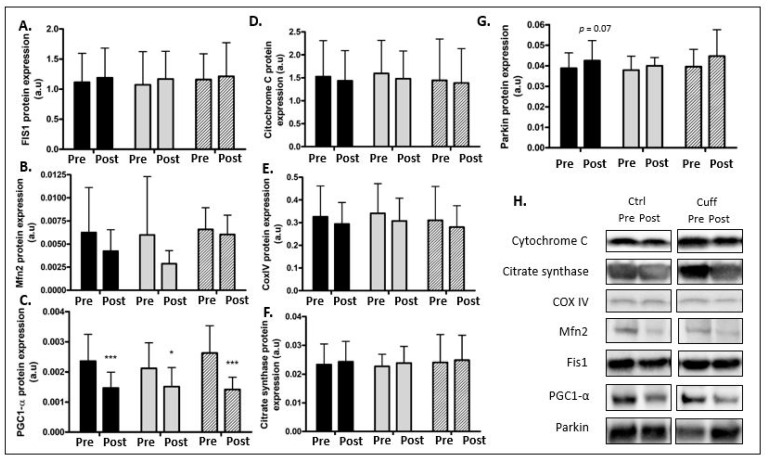
Effect of 5 days of DI on mitochondrial capacity. Mitochondrial content, activity, and biogenesis evaluated by western blot analysis. (**A**) FIS1 protein expression. (**B**) MFN2 protein expression. (**C**) PGC1-α protein expression. (**D**) Cytochrome C protein expression. (**E**) COXIV protein expression. (**F**) Citrate synthase protein expression. (**G**) Parkin protein expression. (**H**) Western blot picture analyzed by groups. * different from pre value in the same group. * *p* < 0.05; *** *p* < 0.001.

**Figure 8 ijms-22-12064-f008:**
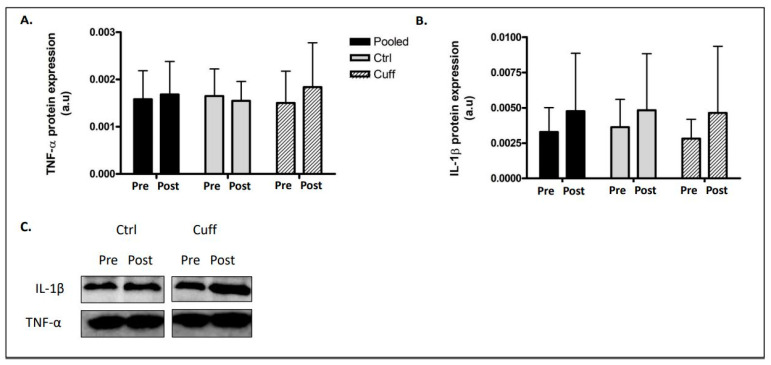
Effect of 5 days of DI on muscle inflammation process. Inflammation activation evaluation by western blot analysis. (**A**) TNF-α protein expression. (**B**) IL-1β protein expression. (**C**) Western blot picture analysed by groups. * different from pre values in the same matched paired group analysis.

**Figure 9 ijms-22-12064-f009:**
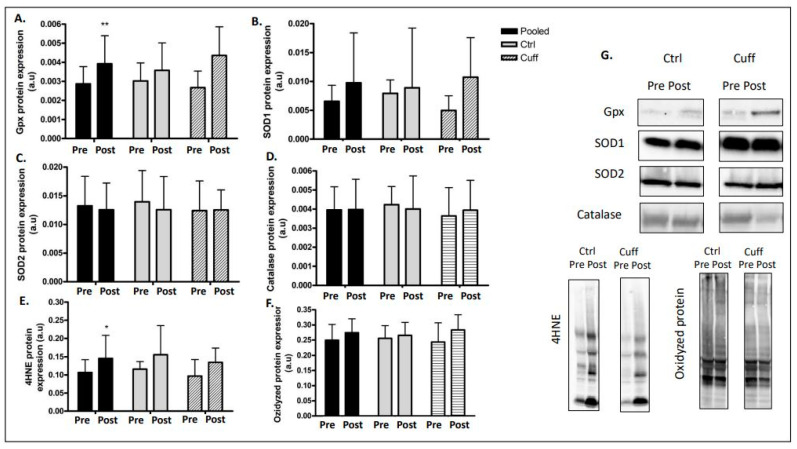
Enzymatic antioxidant defenses and oxidative damages after 5 days of DI. Enzymatic antioxidant defenses access by western blot analysis. (**A**) Gpx protein expression. (**B**) SOD1 protein expression. (**C**) SOD2 protein expression. (**D**) Catalase protein expression. (**E**) 4HNE protein expression. (**F**) Oxydized protein expression. (**G**) Western blot picture analyzed by groups. * different from pre value in the same group. * *p* < 0.05; ** *p* < 0.01.

**Table 1 ijms-22-12064-t001:** Participant characteristics.

	Age (y)	Height (cm)	Weight (kg)	BMI (kg/m^2^)	VO_2_max (mL/min/kg)	Morning HR (bpm)	Morning T (°C)	Morning SBP (mmHg)	Morning DBP (mmHg)
Control (*n* = 9)	33.9 ± 7.1	176 ± 6	73.9 ± 7.5	23.9 ± 1.7	46.5 ± 8.1	57 ± 6	36.4 ± 0.3	115 ± 11	68 ± 5
Cuff (*n* = 9)	34.1 ± 3.7	180 ± 4	74.3 ± 8.8	22.7 ± 1.8	46.9 ± 5.8	58 ± 8	36.4 ± 0.5	117 ± 10	68 ± 9
*p* value	0.93	0.08	0.91	0.16	0.91	0.6	0.71	0.78	0.92
All (*n* = 18)	34.0 ± 5.5	178 ± 6	74.1 ± 8.0	23.3 ± 1.8	46.7 ± 6.9	58 ± 7	36.4 ± 0.4	116 ± 10	68 ± 7

## Data Availability

The data that support the findings of this study are available from the corresponding author, T.B. and A.C., upon reasonable request.

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
