# Peer review of "Early Deconditioning of Human Skeletal Muscles and the Effects of a Thigh Cuff Countermeasure"

_ijms, 2021, doi:10.3390/ijms222112064_

Round 1
Reviewer 1 Report
Review:
Early deconditioning of human skeletal muscles and the effects of a thigh cuff countermeasure
Major comments:
The paper addresses a general question: physiological, histological and molecular changes due to muscle disuse. The authors used thigh cuff to model muscle disuse and compared muscle condition before and after 5 experimental days.
Muscle disuse is a general problem, also among intensive care patients. Yet, the paper is specifically written and focused on the few people that are interested in spaceflight.
It is not convincingly written that the thigh cuff is a good model for muscle disuse.
It is unclear what is the question made by the pooled data. If the question is the effect of thigh cuff on the muscle, the pooled data should be removed.
Figure 1 is at the end – this is confusing.
The authors performed muscle physiology, histology and western blots on bulk protein extract. The experimental design is paired, but it is unclear which statistical tests were paired and which were not. The bar chart presentation does not show the paired design, which is essential in this study.
It is unclear how myofiber typing was made, was it eye-based or based on mean fluorescence intensity? did the authors consider hybrid myofibers? How was the CSA measured? This is crucial to assess if the results in Fig. 3 are correct. The cropped images in Fig. 3 are not informative. Can the authors show the entire cross section? Based on the presented images the CSA is NOT smaller between pre and post.
All western blots missing loading control, without the loading control it is not possible to assess the expression levels. A supplementary material should show all blots.
If the image quantification was correctly done, there is no significant difference between the two conditions, likely due to high variations between subjects.
The markers that were selected for protein synthesis pathway are incorrect: the mTor pathway affects for the ribosome and autophagy (thus both synthesis and breakdown), it is a central regulator of metabolic pathways.
Without a correct presentation of the experiments, conclusions cannot be drawn.
If the results are correct, the study suggests that muscle atrophy and muscle disuse is initiated after 5 days of dry immersion and thigh cuff has no effect.
It is unclear if the conclusion is novel? To my knowledge it is not.
Minor comments:
The term muscle deconditioning is not conventional, either explain (=Lack of muscle fitness)
or replace with a conventional term: muscle weakness.
Abstract:
- “wide range of situations” should be “wide range of conditions”.
- “spaceflight to a sedentary lifestyle and occurs as a result of muscle inactivity”. A more actual example is muscle dysfunction due to disuse for intensive care patients.
- The sentence “To evaluate the mechanisms of early changes and the effect of a thigh cuff (a cardiovascular disorder countermeasure),” is unclear and must be revised as it presents the research question. Do the authors mean thigh cuff muscle, and then they refer to a cardiovascular condition?
- The term “dry immersion” should be explained, why after 5-day muscles were at disused condition?
- “The use of the thigh cuff did not prevent muscle deconditioning.” Is it conclusion or results? Unclear
- The conclusion “These results suggest that the major effects of muscle deconditioning occur during the first few days of inactivity” cannot be made as sampling was made ONLY at one time point during inactivity. The authors can conclude that primary changes of the muscle tissue, due to ….. includes protein changes and muscle histology.
Introduction:
- The introduction starts (first paragraph) with a research question that of interest for a very narrow sector and does not fit to a general journal. As such, either change the research question to broader readers, or submit the paper to a more specialized journal.
- What do you mean “severe events”, last sentence of the introduction? The conditions that you describe are not severe compare with myopathies or dystrophies.
Results:
- In the introduction you wrote 20 men, but in table 1 only 18 men are mentioned.
- Table 1 lacks information: how p-value was calculated, the full name of the abbreviations, covariance: smoking, medications.
- What is MVC?
- All figures: how p-value was calculated.
- Fig. 2A- cuff, the SDs are too large, and the average difference is too small to support statistical significance.
- The definitions of fast and slow fibers are detailed in the Methods, but should also included in the results section to Fig. 2E-F.
The Discussion is too lengthily (written as a review) and should be shortened. Main conclusions should be made instead per experiment or per set of genes. (see suggestion in the major comments section).
Methods:
What is R solution?
Myofiber tying: the specification of the antibodies, including laminin must be included. Both antiMyHC-2A and anti-MyHC-1 are mouse monoclonal, how did you differentiate between them when incubated with a single cross section?
How myofiber CSA was determined, how many myofibers were used to determine mean CSA?
Author Response
We would like to sincerely thank the reviewer for his comments and suggestions, which improved the quality of our manuscript. We (in blue) take back step by step the different comments and recommendations of the reviewer (in black).
Review:
Early deconditioning of human skeletal muscles and the effects of a thigh cuff countermeasure
Major comments:
The paper addresses a general question: physiological, histological and molecular changes due to muscle disuse. The authors used thigh cuff to model muscle disuse and compared muscle condition before and after 5 experimental days. Muscle disuse is a general problem, also among intensive care patients. Yet, the paper is specifically written and focused on the few people that are interested in spaceflight.
Response: We agree with the reviewer. This study was conducted by the French Space Agency, with a specific focus on the space physiology. These results are important for the space domain, and also on earth for the clinical domain. We added several sentences following your comments to mention the clinical applications. Using an original model of disuse and unloading, we were convinced that our study could greatly contribute to the special issue “Skeletal Muscle Molecular Signaling in Various Models of Disuse and Unloading”.
It is not convincingly written that the thigh cuff is a good model for muscle disuse.
Response: The dry immersion was the experiment model for inducing muscle disuse and weakness, and, as in microgravity, it induces a body fluid shift and vascular disorders. In this context, this experiment was designed by the French Space Agency to evaluate the effects of a thigh cuff as body fluid shift countermeasure, and also, to evaluate its effects on the others functions like muscle function.
It is unclear what is the question made by the pooled data. If the question is the effect of thigh cuff on the muscle, the pooled data should be removed.
Response: As hypothesized, the thigh cuff countermeasure had no effect on muscle function. In this context, we took the possibility of carrying out a statistical analysis including a group where all the subjects were grouped together in order to acquire greater statistical power and a better quality of muscle weakness analysis.
Figure 1 is at the end – this is confusing.
Response: Figure 1 has been moved to the beginning of the result section as recommended.
The authors performed muscle physiology, histology and western blots on bulk protein extract. The experimental design is paired, but it is unclear which statistical tests were paired and which were not. The bar chart presentation does not show the paired design, which is essential in this study.
Response: Additional details have been added in the statistical section and in the figure legends in terms of statistical differences. As described in the method, we used matched pair two ways ANOVA, one way for group and a second for time measurement (Pre and Post dry immersion). We chose this type of graph for an optimized presentation of our results.
It is unclear how myofiber typing was made, was it eye-based or based on mean fluorescence intensity? did the authors consider hybrid myofibers? How was the CSA measured? This is crucial to assess if the results in Fig. 3 are correct. The cropped images in Fig. 3 are not informative. Can the authors show the entire cross section? Based on the presented images the CSA is NOT smaller between pre and post.
Response: The analyses were carried out with the help of a threshold and then verified by eye in order to ensure the quality of the results. Hybrid fibers expressing type 1 and 2 MyHC were not presented independently but included in both types of fibers. At 5 days of dry immersion, they are only a slight transition, and the number of hybrid fibers remains very low.
CSA were measured using particle analysis via Fiji software based on a threshold on the laminin signal. This processing was performed in a double-blind fashion and the image set processed three times (each time by a different researcher) to ensure reproducibility of the analysis. The images in Figure 3 are presented to attest the quality of the labeling.
All western blots missing loading control, without the loading control it is not possible to assess the expression levels. A supplementary material should show all blots. If the image quantification was correctly done, there is no significant difference between the two conditions, likely due to high variations between subjects.
Response: The standardization and verification of the protein content of the samples was performed through two steps. The first step consisted in using strictly 50µg of protein in each sample. The second is the use of stain free technology via pre-cast gels equipped with this technology as loading control. Indeed, this technique is validated and allows an efficient verification of the whole protein spectrum deposited rather than using a single reference protein. We have modified the western blot paragraph in the method section to specify the use of stain free as you recommended.
All the unmodified Stain free and western blot images are available in the folder that was sent during the submission process to the editor.
The markers that were selected for protein synthesis pathway are incorrect: the mTOR pathway affects for the ribosome and autophagy (thus both synthesis and breakdown), it is a central regulator of metabolic pathways.
Response: We agree with the reviewer, the PI3K/Akt/mTOR signaling pathway also partly regulates autophagy and metabolism. It also regulates the ubiquitin-proteasome system, interaction with calpains, apoptosis... mTOR is a central actor of all these signals and regulations within the cell. More and more studies highlight the different interactions and feedback between these different pathways.
However, it is clearly established in the literature that the PI3K/Akt/mTOR pathway largely controls protein synthesis. In this context, we have focused on measuring both the activity of mTOR and its targets (Rps6; 4EBP1) which are essentially focused on protein synthesis. It is also well established that an increase in their activity leads to an increase in the synthesis flux. These markers are classically used in the scientific literature and represent today a gold standard.
Without a correct presentation of the experiments, conclusions cannot be drawn. If the results are correct, the study suggests that muscle atrophy and muscle disuse is initiated after 5 days of dry immersion and thigh cuff has no effect. It is unclear if the conclusion is novel? To my knowledge it is not.
Response: This experiment is original because it allows to study muscle weakness on healthy subjects, and dry immersion is more drastic than bed rest experiment. This experiment allowed to the Space Agency first to evaluating the effects of the thigh cuff on body fluid shift, and secondly to evaluate the effects on the general physiology parameters (muscle weakness with our experiments). Our conclusion, both highlighted the safety use of the tigth cuff regarding muscle system, and in another hand, contributed to increase knowledge on the first step of muscle weakness. A large number of study focused on mid- and long-duration experiments, and so we need actually to always precise the first markers and processes of muscle weakness.
Minor comments:
The term muscle deconditioning is not conventional, either explain (=Lack of muscle fitness) or replace with a conventional term: muscle weakness.
Response: We chose the term “muscle deconditioning” because it is often used in the spatial field. We agree with the reviewer that the term “muscle weakness” is more largely employed, either in space, clinical, and aging fields.
For us, the term “muscle deconditioning” seems to be appropriate to well describe the response of skeletal muscle system to a decrease in mechanical constraints, and we would like to keep it.
Abstract:
“wide range of situations” should be “wide range of conditions”.
Response: We made the modification.
“spaceflight to a sedentary lifestyle and occurs as a result of muscle inactivity”. A more actual example is muscle dysfunction due to disuse for intensive care patients.
Response: We agree with the reviewer but, as our experiment focused on healthy subjects, we used the term “inactivity”. In accordance with the reviewer, our study is also very relevant to evaluate muscle weakness associated with intensive care, and so we included an additional sentence at the end of abstract.
The sentence “To evaluate the mechanisms of early changes and the effect of a thigh cuff (a cardiovascular disorder countermeasure),” is unclear and must be revised as it presents the research question. Do the authors mean thigh cuff muscle, and then they refer to a cardiovascular condition?
Response: We suppressed “(a cardiovascular disorder countermeasure),” in our sentence.
The term “dry immersion” should be explained, why after 5-day muscles were at disused condition?
Response: The immersion protocol and its effects on muscle function are discussed in the article. Indeed, the length of the abstract does not allow us to give as much details as we would like. We chose to better present the rest of the study in the abstract than to explain the model (which is relatively well known and explained later).
“The use of the thigh cuff did not prevent muscle deconditioning.” Is it conclusion or results? Unclear
Response: This sentence is a conclusion. We replace it by “The use of the thigh cuff did not prevent muscle deconditioning or impact muscle function”. We moved this sentence to the conclusion part of the abstract as recommended.
The conclusion “These results suggest that the major effects of muscle deconditioning occur during the first few days of inactivity” cannot be made as sampling was made ONLY at one time point during inactivity. The authors can conclude that primary changes of the muscle tissue, due to ….. includes protein changes and muscle histology.
Response: For this conclusion, we rely on both our data and those in the literature. Indeed, the extent of muscular deconditioning following a dry immersion is very close to that during space flights of the same duration. By comparing these data at 5 days with measurements after longer space flights, we realize that a large part of the adaptations, notably in terms of muscle mass on the vastus lateralis is already in place. The literature using animal models also clearly highlight the exponential decrease of muscle mass during the first week (Cros et al, 2009).
Introduction:
The introduction starts (first paragraph) with a research question that of interest for a very narrow sector and does not fit to a general journal. As such, either change the research question to broader readers, or submit the paper to a more specialized journal.
Response: This paper, although strongly attached in its writing to the space domain, was submitted following an invitation for the special issue "Skeletal Muscle Molecular Signaling in Various Models of Disuse and Unloading". We therefore believe that our paper could well contribute to this special issue, by describing an original model inducing drastic muscle weakness. We think that our results are relevant for space, and clinical domains.
What do you mean “severe events”, last sentence of the introduction? The conditions that you describe are not severe compare with myopathies or dystrophies.
Response: “Severe events” refers to a drastic model in healthy subjects, (which involve more decrease of mechanical constraints than bedrest for example), and fast deconditioning. We agree that dystrophies were more severs, but it is not the object of our experiments.
Results:
In the introduction you wrote 20 men, but in table 1 only 18 men are mentioned.
Response: We have corrected this error in the introduction and abstract. Indeed, two of the twenty subjects had to be excluded from the protocol because of allergy problems not known at the time of recruitment.
Table 1 lacks information: how p-value was calculated, the full name of the abbreviations, covariance: smoking, medications.
Response: As the result respect normality, the p-value was calculating with t-test to compare groups with each other.
What is MVC?
Response: The MVC is the maximal isometric voluntary contraction (N.m). We add explanation in results and in method.
All figures: how p-value was calculated.
Response: Differences between pre-DI and post-DI and between groups were evaluated using a two ways analysis of variance (ANOVA) matched-pairs with Tuckey HSD post hoc or Friedman ANOVA when the data deviated from a normal distribution, as determined using the Shapiro–Wilk normality test. Statistical analyses were performed using Statistical Software (version 7.1) and graphs were created using GraphPad Prism4 software (San Diego, USA). Statistical significance was set at p<0.05.
We had the post hoc test in statistical section.
Fig. 2A- cuff, the SDs are too large, and the average difference is too small to support statistical significance.
Response: The statistical analysis has been made strictly and we obtained this result despite a large variability between subjects. Indeed, the variability between subjects is high, but all subjects (matched pairs) loose muscle strength, leading to this statistical difference.
The definitions of fast and slow fibers are detailed in the Methods, but should also included in the results section to Fig. 2E-F.
Response: These results by fiber types were detailed in Result 2.1.
The Discussion is too lengthily (written as a review) and should be shortened. Main conclusions should be made instead per experiment or per set of genes. (see suggestion in the major comments section).
Response: Due to the large panel of results and the link between them, we attached a lot of importance to the discussion. Furthermore, we also compared our results with those of the literature, regarding different model of disuse. As we are not limited in length, we would like to maintain all our discussion.
Methods:
What is R solution?
Response: We detailed the R solution in the method section as recommended
Myofiber tying: the specification of the antibodies, including laminin must be included. Both antiMyHC-2A and anti-MyHC-1 are mouse monoclonal, how did you differentiate between them when incubated with a single cross section?
Response: We included laminin antibody in antibody table. The labelling was made on two different sections (serial section). One section with laminin and MyHC-1 and a second with laminin and MyHC-2.
How myofiber CSA was determined, how many myofibers were used to determine mean CSA?
Response: By mean, 430 muscle fibers were analyzed by section to determine the CSA, depending of muscle biopsy.
Reviewer 2 Report
The manuscript ijms-1391051 describes the early mechanisms of muscle deconditioning following five days of DI, a commonly validated experimental protocol for studying the effects of hypogravity, and evaluated the effects of thigh cuff countermeasure which represent a symptom relief measure used in space exploration and some clinical research.
DI, induces a loss in muscle strength, muscle fibre atrophy, impaired protein balance, and decreased oxidative capacity in both conditions. Thigh cuff countermeasure showed beneficial effects to prevent the adverse effects of body fluid redistribution induced by 5 days of DI.
In particular, attention is paid to the first phase of hypogravity in which there is a drastic reduction of neuromuscular function also due to the increase in atrophic events.
This topic is particularly important as society projects itself into a future in which space travel will be more frequent and therefore the need and attention to safeguard the health of astronauts increases.
Some inconsistencies do not make the manuscript publishable in this version.
The authors can easily resolve these comments.
Major:
a) In many cases, (figures 4-9) there seems to be no correspondence between the histograms and the images of the WB.
A striking example is the IL-1beta expression analysis in which the POST seems to indicate a protein expression of 20-50 times higher than the PRE.
How was the densitometry of the bands determined? On which reference protein?
b) Regarding the methodology, I would kindly ask the authors for an extra comment.
Although the methods used are among the most reliable for this type of study, the fact that there are time intervals between hypogravity and normogravity (necessary for physiological situations and for experimental tests) makes this situation little correspondent with reality of the condition that astronauts live in space.
The authors should kindly comment on this discrepancy.
Minor:
a) Please, put the legend of figure 2 under the graph. In this way the figure can also be enlarged to make it easier to consult.
b) Please, paragraphs 4.5 and 4.6 should be reversed.
c) Please check the abbreviations in the text (i.e. mTor, mTOR).
Author Response
We would like to sincerely thank the reviewer for his comments and suggestions, which improved the quality of our manuscript. We (in blue) take back step by step the different comments and recommendations of the reviewer (in black).
The manuscript ijms-1391051 describes the early mechanisms of muscle deconditioning following five days of DI, a commonly validated experimental protocol for studying the effects of hypogravity, and evaluated the effects of thigh cuff countermeasure which represent a symptom relief measure used in space exploration and some clinical research. DI induces a loss in muscle strength, muscle fibre atrophy, impaired protein balance, and decreased oxidative capacity in both conditions. Thigh cuff countermeasure showed beneficial effects to prevent the adverse effects of body fluid redistribution induced by 5 days of DI. In particular, attention is paid to the first phase of hypogravity in which there is a drastic reduction of neuromuscular function also due to the increase in atrophic events.
This topic is particularly important as society projects itself into a future in which space travel will be more frequent and therefore the need and attention to safeguard the health of astronauts increases. Some inconsistencies do not make the manuscript publishable in this version.
The authors can easily resolve these comments.
Major:
a) In many cases, (figures 4-9) there seems to be no correspondence between the histograms and the images of the WB. A striking example is the IL-1beta expression analysis in which the POST seems to indicate a protein expression of 20-50 times higher than the PRE. How was the densitometry of the bands determined? On which reference protein?
Response:
Concerning the western blot analysis, the histograms and images have been verified. As required, we sent all the uncropped, untouched, full original images of western blots to the editor. The differences in terms of visuals versus graphs are mainly due to the large inter-individual variability present in these human experiments.
Concerning the IL-1b expression, we chose a more representative example of the mean, and put this image in the figure 8.
Finally, concerning the densitometry of the bands, the standardization of the protein content of the samples was performed throughout two steps. The first one consisted in using strictly 50µg of protein in each sample. The second one was to use stain free technology via pre-cast gels equipped with this technology. Indeed, this technique is validated and allows an efficient verification of the whole protein spectrum deposited rather than using a single reference protein. All the untouched and uncropped Stain free images are available in the folder containing the western blot images sent to the editor.
- b) Regarding the methodology, I would kindly ask the authors for an extra comment. Although the methods used are among the most reliable for this type of study, the fact that there are time intervals between hypogravity and normogravity (necessary for physiological situations and for experimental tests) makes this situation little correspondent with reality of the condition that astronauts live in space. The authors should kindly comment on this discrepancy.
Response: The 5-day dry immersion protocol was very strict. Indeed, no test performed during the immersion phase required to leave this position. For practical needs in terms of hygiene, these were carried out on a bed in a reclining position (head down tilt bed rest) after an assisted transfer to avoid any postural reload. The daily toilet in the head down tilt bed rest position didn’t exceed 15min per day. You are entirely right in saying that there are differences between dry immersion and space flight, the first one being that astronauts move unlike immersed subjects. On the other hand, dry immersion is interesting because the pressure surface on the back of the body due to gravity is less important than in a bedrest model. However, the space clinic has attached itself to minimize at most all the disturbances in order to make this protocol as close as possible to the reality of the space flights.
Minor:
a) Please, put the legend of figure 2 under the graph. In this way the figure can also be enlarged to make it easier to consult.
Response: The size and position of the graph legends have been redesigned
- b) Please, paragraphs 4.5 and 4.6 should be reversed.
Response: These two paragraphs have been reversed.
- c) Please check the abbreviations in the text (i.e. mTor, mTOR).
Response: We checked and harmonized the abbreviations and chose the “mTOR” format.
Round 2
Reviewer 1 Report
no suggestions,
the manuscript improvement is limited, and remains of interrest for a limited readers' group.
Author Response
We are aware that this study, conducted under the aegis of our space agency, is mainly part of studies on the effects of microgravity and projects of short or long duration space flights. However, for decades, these models (prolonged bed rest, dry immersion, or unilateral suspension of the leg - ULLS), developed by space sector, have also made it possible to study, in healthy subjects, the consequences of hypoactivity (relative to sedentary lifestyle, clinical bed rest, post-injury, intensive care, ...) and especially to assess prevention methods (exercise, supplementation, …). In this context, these results can contribute to knowledge in the spatial domain, but also in the clinical domain.
Reviewer 2 Report
I thank the authors for taking my comments into consideration and resolving the issues raised.
Minor:
a) I continue to be of the opinion that, although this protocol is even closer to the conditions supported by astronauts, in reality, unfortunately, it still does not reflect the situation that this category of people has to face in its role.
Perhaps the authors should consider a more extensive comment on this issue in the discussion. It should certainly be placed as a limitation even if it is actually among the most reliable protocols for this type of research.
b) the authors should double-check all the captions of the figures as reference is made to statistical significances that are not present in the histograms.
Author Response
We sincerely thank the reviewer for these last comments. We (in blue) take back step by step the different comments of the reviewer (in black).
Minor:
- a) I continue to be of the opinion that, although this protocol is even closer to the conditions supported by astronauts, in reality, unfortunately, it still does not reflect the situation that this category of people has to face in its role.
Perhaps the authors should consider a more extensive comment on this issue in the discussion.It should certainly be placed as a limitation even if it is actually among the most reliable protocols for this type of research.
We agree and have added a sentence at the end of our conclusion :
“This experimental dry immersion model does not mimic the real life conditions of astronauts.
However, it is one of the most reliable protocols for this type of research and is a unique opportunity to develop countermeasures of muscle deconditioning on healthy people.”
- b) the authors should double-check all the captions of the figures as reference is made to statistical significances that are not present in the histograms.
We have carefully double-check the figure captions and correct them.